# The Influence of Hyperthyroidism on the Coagulation and on the Risk of Thrombosis

**DOI:** 10.3390/jcm13061756

**Published:** 2024-03-19

**Authors:** Nebojsa Antonijevic, Dragan Matic, Biljana Beleslin, Danijela Mikovic, Zaklina Lekovic, Marija Marjanovic, Ana Uscumlic, Ljubica Birovljev, Branko Jakovljevic

**Affiliations:** 1Clinic for Cardiology, University Clinical Center of Serbia, 11 000 Belgrade, Serbia; dragan4m@gmail.com (D.M.); lekoviczaklina@gmail.com (Z.L.); marija4m@yahoo.co.uk (M.M.); anauscumlic@gmail.com (A.U.); drljubicabirovljev@gmali.com (L.B.); 2Faculty of Medicine, University of Belgrade, 11 000 Belgrade, Serbia; biljana_beleslin@yahoo.com (B.B.); jakovljevic@dr.com (B.J.); 3Clinic for Endocrinology, University Clinical Center of Serbia, 11 000 Belgrade, Serbia; 4Blood Transfusion Institute of Serbia, 11 000 Belgrade, Serbia; 5Institute for Hygiene and Medical Ecology, 11 000 Belgrade, Serbia

**Keywords:** hyperthyroidism, prothrombotic, risk of thrombosis

## Abstract

**Introduction**: Apart from the well-known fact that hyperthyroidism induces multiple prothrombotic disorders, there is no consensus in clinical practice as to the impact of hyperthyroidism on the risk of thrombosis. The aim of this study was to examine the various hemostatic and immunologic parameters in patients with hyperthyroidism. **Methods**: Our study consists of a total of 200 patients comprised of 64 hyperthyroid patients, 68 hypothyroid patients, and 68 euthyroid controls. Patient thyroid status was determined with standard tests. Detailed hemostatic parameters and cardiolipin antibodies of each patient were determined. **Results**: The values of factor VIII (FVIII), the Von Willebrand factor (vWF), fibrinogen, plasminogen activator inhibitor-1 (PAI-1), and anticardiolipin antibodies of the IgM class were significantly higher in the hyperthyroid patients than in the hypothyroid patients and euthyroid controls. The rate of thromboembolic manifestations was much higher in hyperthyroid patients (6.25%) than in hypo-thyroid patients (2.9%) and euthyroid controls (1.4%). Among hyperthyroid patients with an FVIII value of ≥1.50 U/mL, thrombosis was recorded in 8.3%, while in hyperthyroid patients with FVIII value ≤ 1.50 U/mL the occurrence of thrombosis was not recorded. The incidence of atrial fibrillation (AF) was significantly higher (8.3%) in the hyperthyroid patients compared to the hypothyroid patients (1.5%) and euthyroid controls (0%). **Conclusions**: High levels of FVIII, vWF, fibrinogen, PAI-1, and anticardiolipin antibodies along with other hemostatic factors contribute to the presence of a hypercoaguable state in patients with hyperthyroidism. The risk of occurrence of thrombotic complications is especially pronounced in patients with a level of FVIII exceeding 150% and positive anticardiolipin antibodies of the IgM class. Patients with AF are at particularly high risk of thrombotic complications due to a hyperthyroid prothrombotic milieu.

## 1. Introduction

The significance of the data on hyperthyroidism as a probable prothrombotic condition has not been sufficiently implemented in clinical practice, nor are large studies consistent in asserting the association between elevated thyroid hormone values and thrombotic events [1,2]. The results of a new meta-analysis and the findings of other authors indicate that the prothrombotic milieu induced by hyperthyroidism contributes to the increased risk of venous thromboembolism (OR 1.322, 95% CI: 1.278–1.368) [1]. It is believed that the emergence of a prothrombotic (hypercoagulable and hypofibrinolytic) state in hyperthyroidism is significantly contributed to by an elevated level of factor VIII (FVIII), von Willebrand factor (vWF), fibrinogen, and plasminogen activator inhibitor-1 (PAI-1), the level of which increases gradually with the level of thyroid hormones [2,3,4,5]. Atrial fibrillation (AF)—as a special risk factor for cerebral and peripheral thromboembolism—is registered in 5–15% of patients with hyperthyroidism. Besides the unquestionable position that AF itself is a risk for thromboembolism even in non-thyroid patients, the existence of a prothrombotic state in hyperthyroidism and its correlation with thromboembolic complications is not mentioned in the most important recommendations or marked as controversial (American College of Cardiology/American Heart Association 2014 (ACC/AHA 2014), European Society of Cardiology 2020 (ESC 2020)).

The most important and most frequently applied risk scores for the occurrence of cerebral infarction (stroke) and peripheral thromboembolism (CHADS2-VASc, GARFIELD-AF, ATRIA, ABC-stroke) do not include the existence of hyperthyroidism as a risk factor for the occurrence of thromboembolic events [6,7,8,9].

Considering hemostatic prothrombotic factors in patients with thyroid dysfunction, some authors emphasize the possibility of the influence of elevated levels of anticardiolipin antibodies (aCLA) on the occurrence of a prothrombotic state in hyperthyroid patients [10].

The aim of this study was to examine the various hemostatic and immunologic parameters in patients with hyperthyroidism and compare them with the parameters of the patients with hypothyroidism as well as with the euthyroid controls.

## 2. Materials and Methods

### 2.1. The Studied Population

The study is prospective and included 200 participants. The study was conducted in the University Clinical Center of Serbia, Belgrade, between March 2022 and April 2023.

The diagnosis of hyperthyroidism was established in those patients with decreased TSH levels below the reference range and elevated levels of fT4 [11]. The diagnosis of overt (clinically manifested) hypothyroidism was established in those with elevated TSH levels above the reference range and decreased levels of fT4.

A standard clinical and electrocardiographic examination was performed on all subjects.

The exclusion criteria included the presence of infection, malignancy, moderate to severe renal insufficiency, other significant comorbidities, oral contraceptive therapy, and other medications that could potentially affect the analyses.

### 2.2. Laboratory Analysis

Laboratory analyses were conducted in the morning before meals. Tests for detecting thrombophilia, including antithrombin (AT), protein C (PC), protein S (PS), and lupus anticoagulants (LA), were performed outside the acute thrombotic phase of the disease.

The value of T4 was determined using the radioimmunoassay (RIA) method. Detection of thyroid hormone levels (fT4, TSH) was performed on a gamma counter from LKB 1272 CLINIGAMA. CIS tests were used. The reference range for T4 is 55–160 nmol/L. The fT4 value was determined using the CIS RIA method, with reference values of 7–18 ng/L. The TSH value was determined using the immunoradiometric (IRMA) method, with reference values of 0.15–5.9 mU/L.

The assessment of hemostasis was conducted using standard laboratory tests that included the analysis of prothrombin time (PT), activated partial thromboplastin time (APTT), PC, PS, antithrombin, fibrinogen, and PAI-1. These analyses were performed on a “Dade Behring BCS XP System; Siemens Healthtineers; 91301 Forchheim; Germany” apparatus.

For the determination of coagulation activity of FVIII and vWF activity, a coagulometric method was employed using IL tests on an “Instrumentation Laboratory ACL 6000; Beckman; Brea-California; USA” apparatus.

To detect the presence of lupus anticoagulants, the following tests were used:Activated partial thromboplastin time using IL cephaloplastin reagent, with reference values of 24–36 s. The test was performed on the ACL 6000 apparatus.Dilute Russell’s Viper Venom Time (DRVVT) using IL Test LAC Screen and IL Test LAC Confirm. The test result is calculated as the ratio of LAC Screen to LAC Confirm. Reference value: 0.8 to 1.2; a value higher than 1.2 indicates the presence of lupus anticoagulants [12,13].

Anticardiolipin antibodies were determined using a commercial ELISA test (Binding Site UK) on a spectrophotometer. Reference values for the aCLA IgG class are less than 10 GPL U/mL, and for the aCLA IgM class, values are less than 10 MPL U/mL.

The semiquantitative negativity of anticardiolipin antibodies (0) is defined as a value below the upper limit of the reference range. Semiquantitative weak positivity of anticardiolipin antibodies is defined as a value one to two times higher than the upper reference limit. Semiquantitative strong positivity is defined as a value two times higher than the defined upper limit of normal values. The same methodology was applied for the semiquantitative description of aCLA IgG and IgM classes.

### 2.3. Statistical Analysis

In the first phase of data processing, a database was formed, followed by sorting, grouping, and tabulating the results based on the examined characteristics of both the study and control groups. Categorical variables were presented as absolute values and percentages, and comparisons were made using the χ^2^ test and Fisher’s exact test for probabilities. Continuous variables were presented as mean values with standard deviation (SD). Continuous variables with a normal distribution were compared using the Student’s *t*-test, while those without a normal distribution were compared using the Mann–Whitney U test. To assess the significance of differences among three continuous variables, analysis of variance (ANOVA) was used for variables with a normal distribution, and the Kruskal–Wallis test was used for variables without a normal distribution. A *p*-value less than 0.05 was considered statistically significant. All statistical analyses were conducted using an IBM SPSS Statistics V.20.0 software package.

## 3. Results

The prospective study included 200 participants overall. Of these, 68 were patients with hypothyroidism, 64 patients with hyperthyroidism, and 68 healthy euthyroid participants of equivalent age.

The average age of the entire group (including hypothyroid, hyperthyroid, and euthyroid individuals) was 45.17 years ± 14.86 years. The youngest patient was 16, and the oldest was 83 years old. Although the average age was highest in the hypothyroid group, no statistically significant difference in age was found among the examined groups (*p* = 0.129), Table 1.

The mean body-mass index (BMI) in hypothyroid patients was significantly higher than that in euthyroid participants and hyperthyroid patients. There was no statistically significant difference in mean BMI between hyperthyroid and euthyroid participants—Table 1.

In the total study population, there were 51 males (25.5%) and 149 females (74.5%). In each of the three groups, the female gender was more represented. The difference in gender distribution among the examined groups was statistically highly significant (*p* < 0.001)—Table 1.

The analysis of patients with hypothyroidism, hyperthyroidism, and euthyroid controls in relation to the presence or absence of absolute arrhythmia, and their mutual comparison, is shown in Table 1.

Atrial fibrillation was significantly more common in patients with hyperthyroidism than in patients with hypothyroidism and euthyroid controls (*p* = 0.022). Patients with hyperthyroidism and electrocardiographically documented AF were significantly older (average age 62.2 ± 17.4 years) than patients with hyperthyroidism without absolute arrhythmia (average age 41.63 ± 12.3 years) (*p* = 0.012).

During the follow-up, through the active phase of the disease, thrombotic manifestations occurred in the group of hyperthyroid patients with the highest frequency (6.3%) (one with myocardial infarction, two with cerebrovascular infarction, one with pulmonary embolism) compared to 2.9% of patients with hypothyroidism (two patients with myocardial infarction) and 1.5% of euthyroid patients (one patient with myocardial infarction). Despite the evident differences between the groups, due to the small number of patients with thromboses, a statistically higher frequency of thrombosis in hyperthyroid and hypothyroid patients compared to euthyroid control subjects was not proven.

Among five patients with AF and hyperthyroidism, two had thrombotic episodes (one had a cerebrovascular infarction, and the other had a pulmonary embolism). Three patients with AF and hyperthyroidism did not have thrombotic manifestations. Out of 59 patients with hyperthyroidism without AF, 2 patients had thrombotic episodes.

Semiquantitative analysis of aCLA of the IgG class found that in the group of hypothyroid patients, 89.5% were negative, 5.3% were weakly positive, and 5.3% were strongly positive. In the hyperthyroid patient group, 94.1% of participants were negative, while 5.9% were weakly positive. There were no patients with strong positivity for the aCLA IgG class in the hyperthyroid patient group (0%). In the euthyroid group, all participants were negative for the presence of aCLA IgG (100%). Statistical analysis did not find a significant difference in the examined groups regarding semiquantitative analysis of anticardiolipin antibodies of the IgG class (*p* = 0.113).

Semiquantitative analysis of anticardiolipin antibodies of the IgM class revealed that in the group of hypothyroid patients, 94.4% were negative, 5.6% were strongly positive, and there were no patients with weak positivity (0%). In the hyperthyroid patient group, 62.2% were negative, 18.8% were weakly positive, and 18.8% were strongly positive. In the euthyroid group, all participants were negative for the presence of the aCLA IgM class. Statistical analysis found a significant difference among the examined groups in a semiquantitative analysis of anticardiolipin antibodies of the IgM class (*p* = 0.000).

The values of fibrinogen and natural anticoagulants (antithrombin III, PC, PS) in relation to thyroid status are presented in Table 2 and Figure 1. By comparing fibrinogen values between defined groups, it was found that patients with hyperthyroidism had a significantly higher mean fibrinogen value compared to patients with hypothyroidism and control euthyroid subjects (*p* = 0.000). Additionally, it was established that patients with hyperthyroidism more frequently had a fibrinogen value above 4.1 g/L compared to the euthyroid participant group.

Patients with hyperthyroidism had significantly higher levels of FVIII and vWF compared to those patients with hypothyroidism and euthyroid individuals. The rate of FVIII values over 1.5 U/mL was significantly higher in patients with hyperthyroidism (79.4%) than in patients with hypothyroidism (17.6%) and in euthyroid individuals (2.9%).

In the group of hyperthyroid patients, the mean value of PC was significantly lower compared to the group of hypothyroid patients and the control euthyroid group (*p* = 0.000)—Table 2, Figure 1. The representation of PC values below the lower limit of normal (<69%) was similar among the examined groups.

A significantly lower mean value of PS was found in the group of hyperthyroid patients and the group of euthyroid participants compared to the group with hypothyroidism (*p* = 0.001)—Table 2, Figure 1. The representation of PS values below the lower limit of normal (<65%) was similar among the examined groups.

The mean value of antithrombin III was significantly higher in the group of hyperthyroid patients in comparison to the euthyroid control and group of hypothyroid patients (*p* = 0.001)—Table 2 and Figure 1. Furthermore, patients with hypothyroidism had antithrombin III values below the lower limit of normal (<75%) significantly more frequently than patients with hyperthyroidism and participants from the euthyroid control group.

The values of plasminogen and PAI-1 in relation to thyroid status are presented in Table 2 and Figure 2. Although a higher percentage of patients with hyperthyroidism had plasminogen values below the reference limit (<75%) compared to euthyroid patients and patients with hypothyroidism, this difference was not statistically significant. However, by comparing the mean values of plasminogen, a significantly lower value was found in the group of patients with hyperthyroidism compared to the value in the group of hypothyroid patients and the value in the group of euthyroid participants (*p* = 0.000).

The PAI-1 value above the lower limit of normal (>3.5 U/mL) did not significantly differ among the examined groups. However, by comparing the mean values of PAI-1 in the defined groups, a significantly higher value of this parameter was found in the group of hyperthyroid patients compared to the mean value in the group of hypothyroid patients (*p* < 0.001) and a not-significantly higher mean value compared to euthyroid participants.

## 4. Discussion

### 4.1. Changes in FVIII Values in Hyperthyroid Patients

Patients with hyperthyroidism had significantly higher levels of FVIII and vWF compared to those patients with hypothyroidism and euthyroid individuals. When interpreting the significance of the increase in FVIII values, it should be kept in mind that FVIII, together with FV, is a key procoagulant factor capable of dramatically increasing FIXa activity and catalyzing FX activation in a dose-dependent manner, noting that small changes in FVIII concentration can have a critical impact [14]. Elevated levels of the mentioned coagulation factors become normalized after appropriate thyroid-suppressive therapy [15]. Several studies indicate that patients with hyperthyroidism in comparison to euthyroid controls show a significant increase in the levels of fibrinogen, FVIII, FIX, and vWF [5,15].

In our study, among hyperthyroid patients with an FVIII value of ≥1.50 U/mL, thrombosis was recorded in 8.3%, while in hyperthyroid patients with an FVIII value ≤1.50 U/mL the occurrence of thrombosis was not recorded. The existence of data indicating a 3–6 times higher relative risk of venous thrombosis and the impact on the occurrence of recurrent venous thrombosis in individuals with FVIII values above 1.5 U/mL (over 150%), as well as the role of FVIII in the pathogenesis of atherothrombosis, highlights the significance of these findings in inducing a prothrombotic state, atherothrombotic complications, and venous thromboembolism in patients with hyperthyroidism [16,17]. Several authors observe a positive correlation between the concentration of FVIII and serum T4 [18,19], while others argue that, among all examined coagulation factors (FV, FVII, FVIII, FIX, FXI, FXII), FVIII is the most sensitive to changes in thyroid hormone concentration and plays a crucial role in thrombus formation during the hyperthyroid stage of the disease [19].

### 4.2. Changes in vWF Values in Hyperthyroid Patients

In our study, values of vWF were significantly higher in hyperthyroid patients than in patients with hypothyroidism and euthyroid controls. It is considered that elevated levels of vWF, FVIII, and fibrinogen, along with reduced fibrinolytic activity and decreased levels of plasminogen, contribute to the presence of a hypercoagulable state and a predisposition to thromboembolism and vascular diseases in patients with hyperthyroidis [20].

### 4.3. Changes in the Concentration of Fibrinogen

In our study, patients with hyperthyroidism had significantly higher levels of fibrinogen compared to both the hypothyroid group and the euthyroid subjects. In studies by other authors, the level of fibrinogen, a well-known acute-phase reactant, is higher in patients with hyperthyroidism compared to euthyroid control subjects [21].

Many studies have confirmed the connection between plasma fibrinogen concentration and coronary heart disease, indicating that elevated fibrinogen is an independent predictor of both initial and recurrent coronary events [22,23,24]. Fibrinogen is a risk factor not only for cardiovascular diseases but also for stroke, transient ischemic attack, and mortality in middle-aged men and men older than 65 years (with little evidence suggesting it as a risk factor in older women) [15].

### 4.4. Hyperthyroidism and Dysfunction of the Fibrinolytic System

In our study, hyperthyroid patients had significantly lower levels of plasminogen than hypothyroid patients and euthyroid subjects. Despite the well-known physiological role of plasminogen in the fibrinolysis system and the expectation that defects in plasminogen synthesis reduce clot lysis and contribute to a prothrombotic tendency [25], only a small number of authors find clinical expression of thrombosis in patients with severe hypoplasminogenemia [25,26,27]. Plasminogen deficiency is considered a rare cause of thrombophilia, and as an individual defect, it does not represent a strong thrombotic risk factor [28].

In our study, even though the mean values of PAI-1 in hyperthyroid patients were higher than those of the euthyroid group (4.81 ± 1.56 U/mL), statistical analysis did not reveal a significant difference between these two groups. The activity of the endogenous fibrinolytic system depends on the balance between plasminogen activators and inhibitors, with PAI-1 being the most significant. It is considered that elevated plasma PAI-1 concentrations alongside fibrinolysis suppression and findings of increased plasma levels of FVIII, vWF, fibrinogen, t-PA, and D-dimer lead to an additional risk of the occurrence of myocardial infarction [5,29]. Dysfunction of the fibrinolytic system may also play a role in the pathogenesis of venous thromboembolism. Increased concentrations of PAI-1 are observed in over 40% of patients with venous thromboembolism [30].

### 4.5. Anticardiolipin Antibodies and Thrombosis

Considering hemostatic prothrombotic factors in patients with thyroid dysfunction, some authors emphasize the possibility of the influence of elevated levels of anticardiolipin antibodies on the occurrence of a prothrombotic state in hyperthyroid patients [15]. In our study, patients with hyperthyroidism had a significantly higher rate of positive titer of aCLA in the IgM class compared to hypothyroid patients and euthyroid controls. Some authors consider aCLA of the IgM class, especially when present in low titers, to be nonpathogenic, unlike the findings of IgG or IgA class aCLA, which are persistently found at higher titers in patients with thromboembolic diseases. However, there are studies that also link IgM aCLA to the occurrence of thrombotic events [31]. Despite the possible explanation that elevated levels of IgM class anticardiolipin antibodies represent an epiphenomenon reflecting the immune background of hyperthyroidism, it is also possible that this factor may contribute to the development of a prothrombotic tendency in patients with hyperthyroidism. Elevated levels of anticardiolipin antibodies, whether in high or low titers, are associated with the occurrence of myocardial infarction and cerebrovascular insult, while only high positive titers of anticardiolipin antibodies are associated with the development of deep vein thrombosis [32]. In our study, among patients with hyperthyroidism there was a low rate of positive IgG anticardiolipin antibodies, and no significant differences were found compared to the hypothyroid patients.

### 4.6. Atrial Fibrillation in Patients with Hyperthyroidism

In our study, AF was significantly more common in patients with hyperthyroidism than in patients with hypothyroidism and euthyroid controls. In a Danish registry with over 40,000 patients with hyperthyroidism, as in our study, AF or Afl was registered in 8.3% of patients within 30 days of the hyperthyroidism diagnosis (11). In studies by other authors, AF is recorded in 10–30% of patients with thyrotoxicosis of all age groups [33,34]. The data indicating that our patients with hyperthyroidism and electrocardiographically registered AF were significantly (*p* = 0.012) older (average age 62.2 ± 17.4 years) than patients with hyperthyroidism without AF (average age 41.63 ± 12.3 years) are consistent with well-established facts that the incidence of AF in hyperthyroid individuals increases with age. In certain studies, the incidence of AF in hyperthyroid patients older than 60 years has been estimated to be between 25–45% in these individuals [33,34,35]. Highlighting the data on the danger of thyrotoxic AF in older individuals should not overshadow the risk of complications in younger patients with the same disease. Results from a study involving 3176 adults with hyperthyroidism and 25,408 euthyroid young adults (18–44 years old), monitored over 5 years, indicate a 1.44 times higher risk of ischemic stroke among the thyrotoxic population. Hyperthyroidism is associated with a prothrombotic state and ischemic stroke independently of AF or flutter findings. Undocumented paroxysmal AF may also contribute to embolic phenomena [8,34].

Various studies estimate the incidence of thrombosis in hyperthyroid states ranging from 8% to 40%, with cerebral embolization being the most commonly registered event [8]. Findings from other authors indicate that thyrotoxic AF is associated with an increased risk of embolization compared to non-thyrotoxic AF [33,34,36,37,38]. The rate of thromboembolic episodes in patients with hyperthyroidism varies from 8.5% to 18.1% in the Hurley et al. study [33,35]. Of the overall of thromboembolic episodes, 53% are cerebral embolism [33]. In our study group, two patients (3.1%) experienced cerebral infarctions—one hyperthyroid patient with AF and one patient with hyperthyroidism in a sinus rhythm. Results from certain studies indicate the association between hyperthyroid prothrombotic conditions and ischemic stroke independently of registered atrial tachycardia, despite the possibility that certain episodes of paroxysmal AF remain undiagnosed [8].

Although AF is a known risk factor for cardio-embolism, the presence of the hypercoagulability parameters in hyperthyroidism further contributes to the occurrence of thrombotic complications [39]. In other words, the thromboembolic potential of patients depends not only on their predisposition to thromboembolic complications with AF, but is significantly enhanced by the endogenous prothrombotic biochemical milieu resulting from high levels of thyroid hormones [39]. In addition to the mentioned data on cerebral and peripheral arterial thromboembolism in patients with hyperthyroidism, certain studies also indicate the risk of venous thromboembolism in patients with hyperthyroidism. A large population study of 53,418 patients registers that in patients with hyperthyroidism, the risk of pulmonary embolism is 2.31 times higher than the control group during a 5-year follow-up after adjusting for confounding factors [40]. A meta-analysis that included 15 studies up until October 2022 found an increased risk of venous thromboembolism (VTE) even in patients with subclinical hyperthyroidism (OR 1.33, 95% CI: 1.29–1.38) [1].

## 5. Study Limitations

Our study was conducted on a relatively small number of patients, so it would be advisable to perform the investigation on a larger sample. The presence of AF was detected through clinical examination and routine electrocardiographic findings, so the number of patients with AF might be higher if continuous ECG monitoring were used for detection. Correlations of other coagulation and hemostasis parameters not investigated in this study might reveal additional risks of thromboembolism. The use of more sensitive techniques for the diagnosis of ischemic stroke, such as NMR, and high-quality CT diagnostics could further establish the association between fibrinogen levels, other coagulation factors, and ischemic stroke or other thromboembolic events.

## 6. Conclusions

The results of our study indicate the presence of a prothrombotic milieu, in patients with hyperthyroidism, that could potentially lead to an increased incidence of thrombotic complications. The risk of thromboembolism is not only higher in older individuals but also in younger people with hyperthyroidism. High levels of hemostatic factors can lead to multiple clinical forms of thromboembolism, especially in the active phase of the disease. The risk of occurrence of a thrombotic complication is especially pronounced in patients with levels of FVIII exceeding 150%, as well as in patients with positive anticardiolipin antibodies of the IgM class.

Given the lack of clear evidence, despite the fact that ACC/AHA from 2006 classified hyperthyroidism as a moderate risk factor, and considering the omission of this classification in later recommendations, due to a larger number of studies and congruent data indicating a prothrombotic tendency induced by hyperthyroidism, it seems reasonable to recommend starting anticoagulant therapy along with adequate thyrostatic therapy when there are no contraindications. New evidence-based studies would be needed to clarify this clinically important issue.

## Figures and Tables

**Figure 1 jcm-13-01756-f001:**
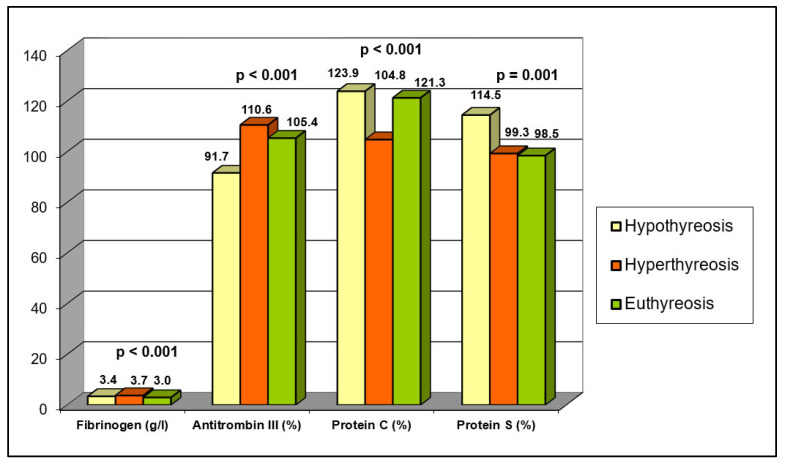
Fibrinogen and Natural Anticoagulant Values in Relation to Thyroid Status.

**Figure 2 jcm-13-01756-f002:**
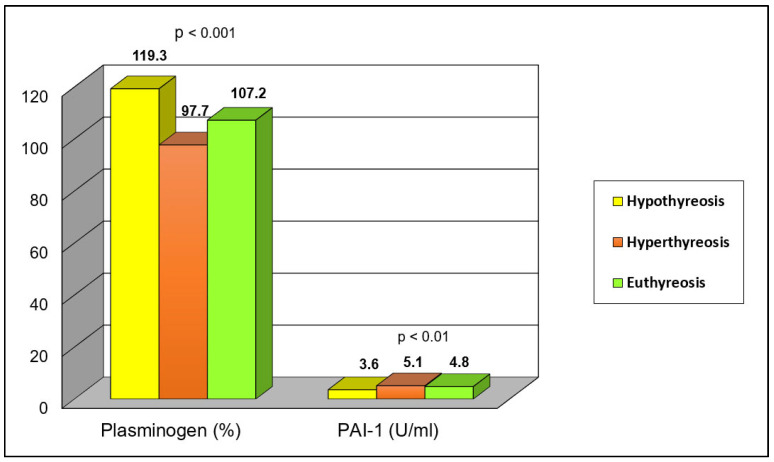
Fibrinolytic System Parameter Values in Relation to Thyroid Status.

**Table 1 jcm-13-01756-t001:** Clinical characteristics and immunologic parameters of patients in relation to thyroid status.

Variable	Hyperthyroidism *n* = 64	Hypothyroidism *n* = 68	Euthyreosis *n* = 68	*p*
Age, years ± SD	43.23 ± 13.77	48.09 ± 13.63	44.06 ± 14.86	0.129
BMI, kg/m^2^ ± SD	23.16 ± 3.74	27.21 ± 6.08	24.37 ± 3.84	>0.05
Female, %	82.8	85.3	55.9	0.000
Atrial fibrillation, %	8.3	1.5	0.0	0.022
Thrombotic events, %	6.3	2.6	1.5	0.313
aCLA IgG p.o., %	5.9	10.6	0.0	0.113
aCLA IgG p.s., %	0.0	5.3	0.0	0.113
aCLA IgG p.m., %	5.9	5.3	0.0	0.113
aLCA IgM p.o., %	37.6	5.6	0.0	0.000
aLCA IgM p.s., %	18.8	5.6	0.0	0.000
aCLA IgM p.m., %	18.8	0.0	0.0	0.000

BMI—body-mass index, aCLA—anticardiolipin antibodies, p.o.—positive overall, p.s.—positive strong, p.m.—positive mildly.

**Table 2 jcm-13-01756-t002:** Hemostatic parameter characteristics in relation to thyroid status.

Variable	Hyperthyroidism *n* = 64	Hypothyroidism *n* = 68	Euthyreosis *n* = 68	*p*
Fibrinogen (g/L), mean ± SD	3.74 ± 0.79	3.41 ± 1.06	2.96 ± 0.74	0.000
Antithrombin III (%), mean ± SD	110.60 ± 14.50	91.65 ± 16.68	105.41 ± 10.67	0.000
FVIII (U/mL), mean ± SD	1.67 ± 0.78	0.93 ± 0.40	0.95 ± 0.24	0.000
FVIII (U/mL) ≥ 1.5 U/mL, %	79.4	17.6	2.9	0.000
vWF (%), mean ± SD	115.8 ± 20.3	81.6 ± 19.6	91.0 ± 12.8	0.000
Protein C (%), mean ± SD	104.84 ± 24.10	123.91 ± 22.90	121.30 ± 21.45	0.000
Protein S (%), mean ± SD	99.33 ± 20.88	114.52 ± 21.65	98.48 ± 26.26	0.001
Plasminogen (%), mean ± SD	97.67 ± 20.39	119.28 ± 19.28	107.23 ± 18.51	0.000
PAI-1 (U/mL), mean ± SD	5.08 ± 1.94	3.62 ± 2.13	4.81 ± 1.56	0.001
Fibrinogen > 4 g/L, %	30.4	18.3	10.6	0.022
Antithrombin > 75%, %	98.1	86.5	100.0	0.001
Protein C < 69%, %	2.0	0.0	0.0	0.326
Protein S < 65%, %	4.8	2.3	9.7	0.266
Plasminogen > 75%, %	90.4	100.0	97.0	0.068
PAI-1 > 3.5 U/mL, %	73.0	53.6	73.8	0.133

FVIII—factor VIII, vWF—Von Willebrand factor, PAI-1—plasminogen activator inhibitor-1.

## Data Availability

The data presented in this study are available on request from the corresponding author.

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
