# Peer review of "The Influence of Hyperthyroidism on the Coagulation and on the Risk of Thrombosis"

_jcm, 2024, doi:10.3390/jcm13061756_

Round 1

Reviewer 1 Report

Comments and Suggestions for Authors

In their study, Antonijevic et al aimed to evaluate whether an augmented prothrombotic milieu in individuals with hyperthyroidism and atrial fibrillation (AF) is correlated with heightened thrombotic complications compared to individuals with hyperthyroidism without AF.

The subject matter of this study is engaging, and the statistical analysis is robust. However, there are certain aspects of the study that necessitate further elucidation by the authors.

  1. Given that the primary objective was to assess the influence of various homeostatic parameters in patients with and without atrial fibrillation (AF), it would have been more appropriate for the authors to establish two distinct groups (A = patients with hyperthyroidism and AF, B = patients with hyperthyroidism without AF). As it stands, the authors conducted various comparisons between patients with hyperthyroidism and hypothyroidism/normal  thyroid function thus confusing the reader.
  2. It is recommended that the authors narrow their focus to the analysis of patients with hyperthyroidism with/without AF, excluding all other patients from this manuscript to ensure a clearer objective.
  3. The discussion section should be condensed. Some portions lack cohesion and are overly verbose. It would be preferable to discuss the results of all prothrombotic parameters in 1-2 paragraphs. Additionally, Section 4.9 should be abbreviated.
  4. In the results section, the authors note that among the 5 patients with hyperthyroidism and AF, only 2 experienced thrombotic complications, including one patient with pulmonary embolism. However, AF and pulmonary embolism are not causally related. The authors should have only included arterial thrombotic episodes as complications due to AF.
  5. The conclusion section contains several overstatements that significantly diminish the quality of the manuscript. The study's aim is not accurately reflected by the statements in lines 729-734. To recommend anticoagulation therapy in patients with hyperthyroidism and a prothrombotic tendency, proper randomized control studies should be conducted. The authors' proposal, not aligned with the study's aim (no mention of AF in lines 729-734), confuses the reader and exaggerates the results.
  6. Numerous typos are present throughout the manuscript, and several abbreviations, such as AF, are not consistently used. These issues should be addressed for greater clarity and professionalism.

Author Response

Point-by-point response to the reviewer’s comments is uploaded as the Word file.

Reviewer 2 Report

Comments and Suggestions for Authors

The abstract is too long and hard to follow.

All abbreviations should be defined at first appearance in the text (ex. BMI line 162).

There is no correlation between the aim of the study and the Material and Methods since the aim of this study, mentioned in the abstract, was to examine the influence of various hemostatic parameters in patients with hyperthyroidism with or without atrial fibrillation, but in the Material and methods mentioned that "The study is prospective and included 200 participants: 51 males and 149 females. 94 The study involved 64 patients diagnosed with hyperthyroidism, 68 patients diagnosed 95 with hypothyroidism, and 68 euthyroid participants who formed the control group (with 96 TSH, fT4 values within reference ranges)"

Results of the study should be presented in the results section (ex lines 94-97 similar to 155-156), not in Material and Methods.

Comments on the Quality of English Language

Major English revision is recommanded. 

Author Response

(The authors gave the same response as above.)

Reviewer 3 Report

Comments and Suggestions for Authors

The submitted paper prospectively explores the associations between dysthiroidism and coagulative status. The manuscript is overall well written and straightforward. I would be more inclusive of results obtained in the title. Please find below specific minor comments:

- please use abbreviations consistently throughout the text

- it appears unclear when and where patients have been enrolled in the study, please specify

- please revise the format of tables, text is inappropriately large and content not adapted to cell dimensions

Author Response

(The authors gave the same response as above.)

Round 2

Reviewer 2 Report

Comments and Suggestions for Authors

I do appreciate the improvements the authors brought to the manuscript, but still, the abstract is too long.

Author Response

The second response to reviewer 2 is located in the attachment. 
